# Development of a red fluorescent protein-based cGMP indicator applicable for live-cell imaging

Mai Takizawa [1,4], Yuri Osuga[1,4], Rika Ishida[1,4], Marie Mita[1,3], Kazuki Harada[1], Hiroshi Ueda [2], Tetsuya Kitaguchi[2 ✉] & Takashi Tsuboi [1 ✉]

Cyclic guanosine 3′, 5′-monophosphate (cGMP) is a second messenger that regulates a variety of physiological processes. Here, we develop a red fluorescent protein-based cGMP indicator, "Red cGull". The fluorescence intensity of Red cGull increase more than sixfold in response to cGMP. The features of this indicator include an $EC_{50}$ of 0.33 μM for cGMP, an excitation and emission peak at 567 nm and 591 nm, respectively. Live-cell imaging analysis reveal the utility of Red cGull for dual-colour imaging and its ability to be used in conjunction with optogenetics tools. Using enteroendocrine cell lines, Red cGull detects an increase in cGMP following the application of L-arginine. An increase in intracellular cGMP is found to be inhibited by $Ca^{2+}$, and L-arginine-mediated hormone secretion is not potentiated. We propose that Red cGull will facilitate future research in cell signalling in relation to cGMP and its interplay with other signalling molecules.

[1] Department of Life Sciences, Graduate School of Arts and Sciences, The University of Tokyo, 3-8-1 Komaba, Meguro, Tokyo 153-8902, Japan. [2] Laboratory for Chemistry and Life Science, Institute of Innovative Research, Tokyo Institute of Technology, 4259 Nagatsuta-cho, Midori-ku, Yokohama, Kanagawa 226-8503, Japan. [3]Present address: Biomedical Research Institute, National Institute of Advanced Industrial Science and Technology, 1-8-31 Midorigaoka, Ikeda, Osaka 563-5577, Japan. [4]These authors contributed equally: Mai Takizawa, Yuri Osuga, Rika Ishida. ✉email: kitaguct-gfp@umin.ac.jp; takatsuboi@bio.c.u-tokyo.ac.jp

Cyclic guanosine 3′, 5′-monophosphate (cGMP) is an important second messenger involved in a variety of physiological functions such as smooth muscle relaxation and phototransduction[1]. Intracellular cGMP concentration is strictly regulated via production by cytosolic/soluble guanylyl cyclase (sGC) or membrane-bound/particulate guanylyl cyclase and through degradation by phosphodiesterase[1]. The complexity of intracellular cGMP dynamics depends on the balance of the activation and expression level of these proteins. Thus, revealing the precise spatiotemporal regulation mechanisms of intracellular cGMP dynamics, particularly its interactions with other signalling molecules such as $Ca^{2+}$, is an important research goal.

sGC is the only conclusively proven receptor for nitric oxide (NO), a signalling molecule produced by the enzyme nitric oxide synthase (NOS) from the amino acid L-arginine[2]. The NO/cGMP signalling pathway is associated with several cell functions, including relaxation of vascular muscles and neurotransmission[3]. Furthermore, evidence suggests that this pathway is involved in exocytotic functions through the activation of some targeted proteins, such as phosphodiesterase, protein kinases, or ionic channels; NO donors, nitrites, and cGMP analogues potentiate glucose-induced insulin secretion from pancreatic β cells[4–6]. Glucagon-like peptide-1 (GLP-1), a gut hormone secreted from enteroendocrine L-cells, plays a role in potentiating glucose-dependent insulin secretion and reducing appetite[7,8]. Glucose, several types of amino acids, and fatty acids have all been identified as secretagogues of GLP-1[9–11]. Recently, it was reported that administration of L-arginine enhances the secretion of GLP-1 and peptide YY (PYY), another gut hormone secreted from L-cells, in both rodents and humans[12,13]. However, the involvement of the NO/cGMP signalling pathway in L-arginine-induced GLP-1 secretion remains to be verified.

Genetically encoded fluorescent protein (FP) indicators based on variants of the green fluorescent protein (GFP) are powerful tools for imaging intracellular cGMP dynamics. Most currently available cGMP indicators are Förster resonance energy transfer (FRET)-based indicators[14,15]. Since FRET imaging requires the acquisition of emitted light from the donor and acceptor proteins, this process has limited flexibility in multicolour imaging. Single FP-based indicators can be used to overcome this problem; thus, many single FP-based indicators have been developed to detect signalling molecules ($Ca^{2+}$ and cAMP) and metabolites (glucose, pyruvate or lactate), including the GFP-based cGMP indicator that we and another group previously developed[16–20].

Optogenetic methods are also useful for modulating intracellular signalling molecules by light. Previously, blue light-activated guanylyl cyclase (bacterial light-activated guanylyl cyclase: BlgC) was purified and engineered onto the base of bacterial light-activated adenylyl cyclase (BlaC) from the *Beggiatoa sp.* PS genome; it has been proposed for use in in vitro experiments involving mammalian cells after human-codon optimisation[21,22]. To extend the colour palette of single FP-based cGMP indicators and enable integration into optogenetic experiments, a nongreen sensor is required.

In the present study, we demonstrate the development of a red single FP (mApple)-based cGMP indicator, namely Red cGull (<u>Red</u> <u>c</u>GMP vis<u>u</u>alising <u>f</u>luorescent protein). We show that live-cell imaging can be conducted using Red cGull under single and dual-colour acquisition and that this indicator can be implemented in optogenetics. We also reveal the $Ca^{2+}$ and cGMP dynamics in L-arginine-induced GLP-1 secretion from enteroendocrine cells.

## Results

### Red cGull is a single FP-based cGMP indicator.
The green FP-based cGMP indicator, Green cGull, was originally reported by our group. It was designed by inserting the cGMP-

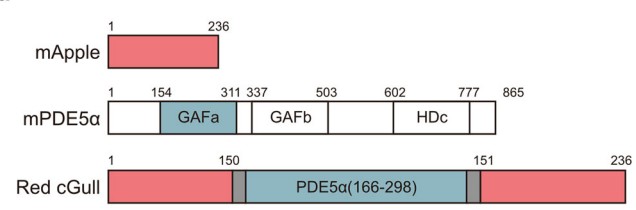

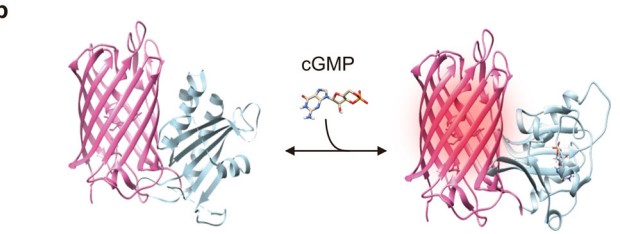

**Fig. 1 Schematic representations of mApple, mPDE5α, and Red cGull. a** Diagrams of mApple, mPDE5α and Red cGull. mPDE5α contains GAF domains for cGMP binding and HD domains for phosphodiesterase activity. See also Fig. S1. **b** Three-dimensional schematic images of Red cGull unbound (left) and bound (right) to cGMP. Images were created using structural data for mCherry (PDB_4ZIO) and PDE5α (cGMP-unbound: PDB_3MF0; cGMP-bound: PDB_2K31).

binding domain of cGMP-specific mouse phosphodiesterase 5α (mPDE5α) near the chromophore of a GFP variant, Citrine, with linker sequences derived from the leucine zipper sequences[22–24]. Here, we inserted the same cGMP-binding domain into a red FP, mApple, to develop a prototype of a red cGMP indicator (Fig. 1a and Supplementary Fig. 1a). Based on our previous work, we learned that mutations in linker peptide sequences largely affect the dynamic range ($F/F_0$) of FI of indicators in response to target molecules. Therefore, we constructed numerous variants of the indicator with different linker lengths and amino acid sequences (Supplementary Fig. 1a, b). Following the screening, we identified the mutant with the largest FI in the presence of 10 μM cGMP, which we named Red cGull (Fig. 1b and Supplementary Fig. 2).

### Red cGull shows a sixfold FI increase in response to cGMP in vitro.
We first analysed the in vitro properties of Red cGull using purified recombinant proteins from *E. coli*. According to fluorescence spectra, Red cGull had excitation and emission peaks at 567 and 591 nm, respectively (Fig. 2a). The FI of Red cGull increased 6.7-fold in the presence of 10 μM cGMP, which is a saturating dose. The absorption spectra of Red cGull showed a higher peak near 450 nm in absence of cGMP, and a higher peak near 570 nm in the presence of 100 μM cGMP (Fig. 2b). Extinction coefficients were calculated from the peak at 562 nm (Table 1). The quantum yield of Red cGull increased 1.5 times in the cGMP saturated state (Table 1). For negative control, Red cGull nega (Supplementary Fig. 1), which showed little change in fluorescence intensity in the presence of cGMP (Supplementary Fig. 3), was selected, and extinction coefficient and quantum yield were displayed in Table 1.

We also examined the dose-response relationship of Red cGull for cGMP and calculated the half-maximal effective concentration ($EC_{50}$) value by fitting a four-parameter logistic curve. The $EC_{50}$ value of Red cGull for cGMP was 0.33 μM; in contrast, Red cGull exhibited a negligible response to a structurally similar substance, cAMP (Fig. 2c). In addition, the response of Red cGull was reversible, and the FI of Red cGull reached 80% of its maximum within 5 s (Supplementary Fig. 4). Taken together, these results indicate that Red cGull detects cGMP and shows a consequent increase in FI in vitro.

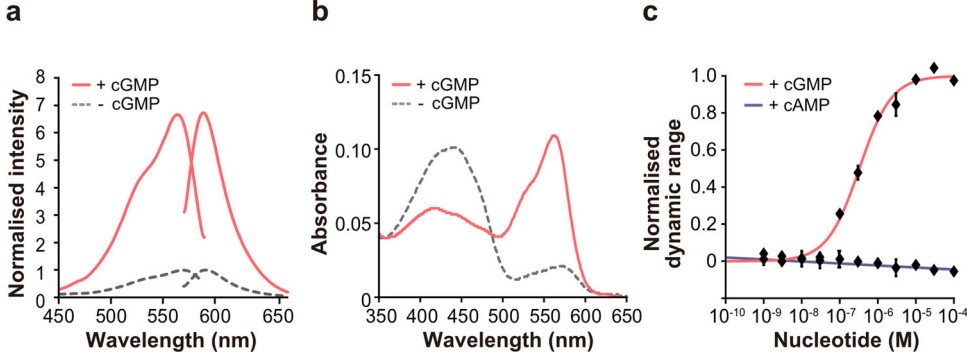

**Fig. 2 Spectral properties of Red cGull. a** The excitation and emission spectra of 5 μM Red cGull in the presence (solid line) and absence (dashed line) of 10 μM of cGMP. The FI was normalised to the maximum FI in the absence of cGMP. **b** The absorption spectra of 20 μM Red cGull in the presence (solid line) and absence (dashed line) of 100 μM of cGMP. The absorption peak of Red cGull decreased to near 450 nm and increased near 570 nm in the presence of cGMP. **c** Dose-response curve of 1 μM Red cGull to cGMP (pink line) and cAMP (purple line). Data were shown as means ± standard deviation ($n = 3$). FI was normalised to the upper and lower asymptotes. The $EC_{50}$ value was calculated from the four-parameter logistic curve as 0.33 μM.

| Table 1 Quantum yields and extinction coefficients of Red cGull and Red cGull nega. | | | | |
|---|---|---|---|---|
| | Quantum yield (-cGMP) | Quantum yield (+cGMP) | Extinction coefficient ($M^{-1}cm^{-1}$) (-cGMP) | Extinction coefficient ($M^{-1}cm^{-1}$) (+cGMP) |
| Red cGull | 0.149 ± 0.007 | 0.225 ± 0.008 | 988 ± 54.5 | 5,638 ± 300.8 |
| Red cGull nega | 0.118 ± 0.005 | 0.116 ± 0.003 | 313 ± 65.0 | 288 ± 89.3 |

The quantum yields were measured by the absolute photoluminescence quantum yield measurement system (Hamamatsu Photonics, C9920-02). Data were shown as means ± standard deviation ($n = 3$). The extinction coefficients were calculated using absorbance at 562 nm (Fig. 2b). Data were shown as means ± standard deviation ($n = 4$).

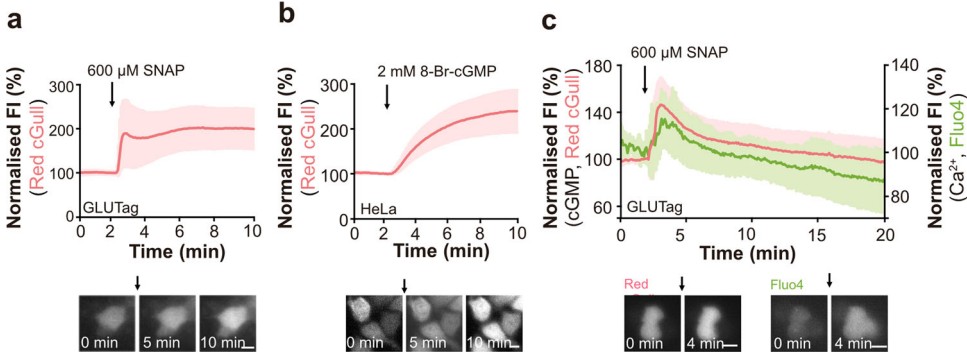

**Fig. 3 Visualisation of intracellular cGMP dynamics by Red cGull. a** Time course of FI (top) and sequential images (bottom) of GLUTag cells expressing Red cGull during the application of 600 μM SNAP. Data were means (line) ± standard deviation (s.d.) (shaded area) ($n = 17$ cells from three independent experiments). The scale bar represents 10 μm. **b** Time course of FI (top) and sequential images (bottom) of HeLa cells expressing Red cGull during the application of 2 mM 8-Br-cGMP. Data are means ± s.d. ($n = 21$ cells from four independent experiments). The scale bar represents 10 μm. **c** Time courses of FI of Red cGull (pink line) and Fluo4 (green line) and sequential images (bottom) of Red cGull-expressed and Fluo4-loaded GLUTag cells during the application of 600 μM SNAP. Data were means ± s.d. ($n = 12$ cells from three independent experiments). Scale bars represent 20 μm.

**Red cGull monitors intracellular cGMP dynamics.** We next validated the utility of Red cGull in live cells. We expressed Red cGull in the mouse enteroendocrine L cell line GLUTag and human cervical epithelial carcinoma cell line HeLa, and then applied cGMP-inducing stimuli, SNAP (NO donor) or 8-Br-cGMP (a membrane-permeable cGMP analogue), to monitor the intracellular cGMP dynamics in modified Ringer's buffer (RB). Application of various concentrations of SNAP or 8-Br-cGMP elicited increases in the FI of Red cGull in each cell line (Fig. 3a, b and Supplementary Fig. 5a, b). For SNAP application, we applied 6 to 600 μM SNAP to GLUTag cells. About 6 μM SNAP induced a small increase in FI, and 60 μM SNAP induced a large increase of FI as 600 μM SNAP but decreased over time. For 8-Br cGMP application, we applied 0.5 to 2 mM 8-Br-cGMP to HeLa cells.

About 0.5 mM 8-Br-cGMP did not induce changes in FI, while 1 mM induced a comparable increase as 2 mM 8-Br-cGMP. For validation of those response, we used Red cGull nega (Supplementary Fig. 5c, d) and δ-FlincG, a previously developed fluorescent protein-based cGMP indicator (means ± standard deviation (s.d.), 113.7 ± 8.3, Supplementary Fig. 6a, c)[20]. $Ca^{2+}$ is an important second messenger and the interplay between $Ca^{2+}$ and cGMP is crucial for many physiological events such as smooth muscle relaxation and intestinal cell proliferation[25]. Given the advantage of single FP-based indicators, which are applicable to dual-colour imaging, we attempted to visualise the changes in both intracellular cGMP and $Ca^{2+}$ concentrations ($[cGMP]_i$ and $[Ca^{2+}]_i$, respectively) by using Red cGull and the green $Ca^{2+}$-sensitive dye Fluo4 in single cells. Application of

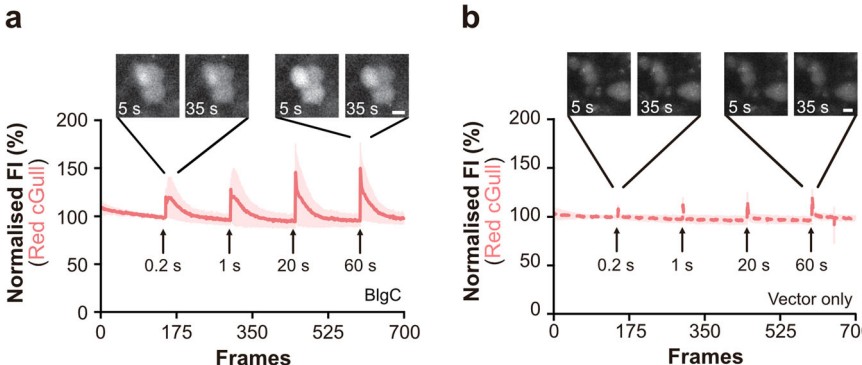

**Fig. 4 cGMP imaging using Red cGull with photoactivated guanylyl cyclase. a**, **b** Sequential images (top) and time courses of the FI (bottom) of GLUTag cells coexpressing Red cGull with photoactivated guanylyl cyclase (BlgC; solid line, **a**) or the vector only (dashed line, **b**) upon blue light laser excitation at 1.2 mW/cm². Data were means (line) ± standard deviation (shaded area) (*a*: *n* = 46 cells from five independent experiments, **b**: *n* = 26 cells from four independent experiments). Scale bars represent 10 μm. Images were taken at one frame per second.

SNAP triggered a transient increase in the FI of both Red cGull and Fluo4 in GLUTag cells (Fig. 3c). These results suggest that Red cGull can be used to monitor intracellular cGMP dynamics in different types of cells and that it is applicable to dual-colour imaging for investigation of the interplay among several signalling molecules.

**Red cGull detects cGMP produced by photoactivated guanylyl cyclase.** To examine whether Red cGull was able to employ in conjunction with optogenetic tools, we coexpressed BlgC and Red cGull in GLUTag cells[21]. By intermittent stimulation with a 1.2 mW/cm² blue light laser (488 nm), the FI of Red cGull was shown to rapidly increase after each stimulus in coexpressing cells (Fig. 4a). In contrast, in a control experiment, blue light laser stimulation itself induced only minor increases in the FI of cells expressing Red cGull and the vector only (Fig. 4b). These results suggest that Red cGull detects cGMP levels produced by photo-activated guanylyl cyclase.

**L-arginine induces cGMP production in the enteroendocrine cell line STC-1.** As previous studies have shown that L-arginine potentiates the secretion of GLP-1 in rodents and humans[12,13], we investigated whether cGMP plays a role in the secretion of gut hormones. First, we used real-time PCR to analyse the mRNA expression of sGC subunits in STC-1 cells, which secrete a variety of gut hormones, including GLP-1, similar to native enteroendocrine cells and are routinely used in in vitro experiments (Fig. 5a)[26]. Real-time PCR analysis revealed mRNA expression of sGC α1 and β1 subunits, which are physiologically functional subunits (Fig. 5a)[27]. Next, we examined the signalling cascades induced by L-arginine in STC-1 cells. Application of L-arginine to Red cGull-expressing STC-1 cells induced an increase in the FI of Red cGull (means ± s.d., 118.4 ± 10.3, *p* < 0.0001, Fig. 5b, d, means ± s.d., 112.7 ± 5.9, *p* < 0.0001, Fig. 5c,). We also applied L-arginine to δ-FlincG-expressing STC-1 cells, and observed the increase of FI as well (means ± s.d., 118.5 ± 12.8, *p* < 0.0001, Supplementary Fig. 6b, c)[20]. L-NAME (a NOS inhibitor) and LY-83583 (a sGC inhibitor) inhibited the L-arginine-induced FI increase of Red cGull, suggesting that cGMP was produced via NOS and sGC activation (L-NAME: means ± s.d., 110.4 ± 5.3, *p* = 0.0419, Fig. 5c, LY-83583: means ± s.d., 112.7 ± 4.2, *p* = 0.0066, Fig. 5d). In a previous study, L-arginine was also sensed by nutrient-sensing receptors including the calcium-sensing receptor (CaSR), G protein-coupled receptor family C subtype A (GPRC6A), and taste 1 receptor member 1/member 3 (T1R1/T1R3), and the activation of CaSR and GPRC6A induced

an increase in $[Ca^{2+}]_i$[28]. In the present study, real-time PCR analysis revealed the mRNA expression of these receptors in STC-1 cells (Supplementary Fig. 7a). Fluo4-loaded STC-1 cells showed an instant increase in FI with the application of L-arginine (means ± s.d., 127.5 ± 26.1, *p* < 0.0001, Supplementary Fig. 7b, c). Co-application of Calhex231 (a CaSR antagonist) and YM-254890 (a $G_q$ protein inhibitor) with L-arginine suppressed the FI increase of Fluo4, whereas Calindol (a GPRC6A antagonist) did not have this effect (Calhex: means ± s.d., 115.8 ± 15.6, *p* < 0.0001, YM-254890: means ± s.d., 117.8 ± 14.8, *p* = 0.0257, Calindol: means ± s.d., 125.6 ± 19.2, *p* > 0.9999, Supplementary Fig. 7c). These results suggest that CaSR and Gq protein play primary roles in L-arginine-induced Ca²⁺ signalling. We also explored the interplay between cGMP and Ca²⁺ in STC-1 cells. Inhibition of $G_q$ protein by YM-254890 increased the FI of Red cGull (means ± s.d., 112.6 ± 5.7, *p* = 0.0005, Fig. 6a), whereas inhibition of cGMP synthesis by LY-83583 had little effect on the FI of Fluo4 (means ± s.d., 139.3 ± 26.4, *p* > 0.9999, Fig. 6b). Those suggest that cGMP production is suppressed by Ca²⁺ and does not affect Ca²⁺ elevation during the application of L-arginine in STC-1 cells. Finally, we investigated the involvement of cGMP on GLP-1 secretion via ELISA. Application of L-arginine significantly increased GLP-1 secretion (means ± s.d., 1.291 ± 0.2, *p* = 0.0103, Fig. 6c), but inhibition of cGMP synthesis had little effect on L-arginine-induced GLP-1 secretion (means ± s.d., 1.423 ± 0.1, *p* = 0.5639, Fig. 6c). On the other hand, inhibition of the $[Ca^{2+}]_i$ increase significantly suppressed GLP-1 secretion (means ± s.d., 0.1688 ± 0.1, *p* < 0.0001, Fig. 6c). Taken together, these findings indicate that L-arginine induces an increase in both $[cGMP]_i$ and $[Ca^{2+}]_i$ by activating the NO/cGMP signalling pathway and CaSR, respectively, in STC-1 cells, but that only Ca²⁺ functions as a potential modulator of GLP-1 secretion (Supplementary Fig. 8).

**Discussion**
Based on our previous method for establishing single FP-based indicators, we developed a single red FP-based cGMP indicator, Red cGull, for use in live-cell imaging. Expanding the colour palette of single FP indicators is important to the improvement of intracellular imaging, taking such imaging from a monochromatic to a multichromatic endeavour. Therefore, red single FP indicators have previously been developed, as exemplified by Ca²⁺, ATP, cAMP and glucose[17,29–31]. However, until now, a red single FP-based indicator for cGMP had not been developed. Red cGull overcomes the limitations of cGMP imaging and is usable in conjunction with optogenetic tools and enables dual-colour imaging with different molecules in a single cell.

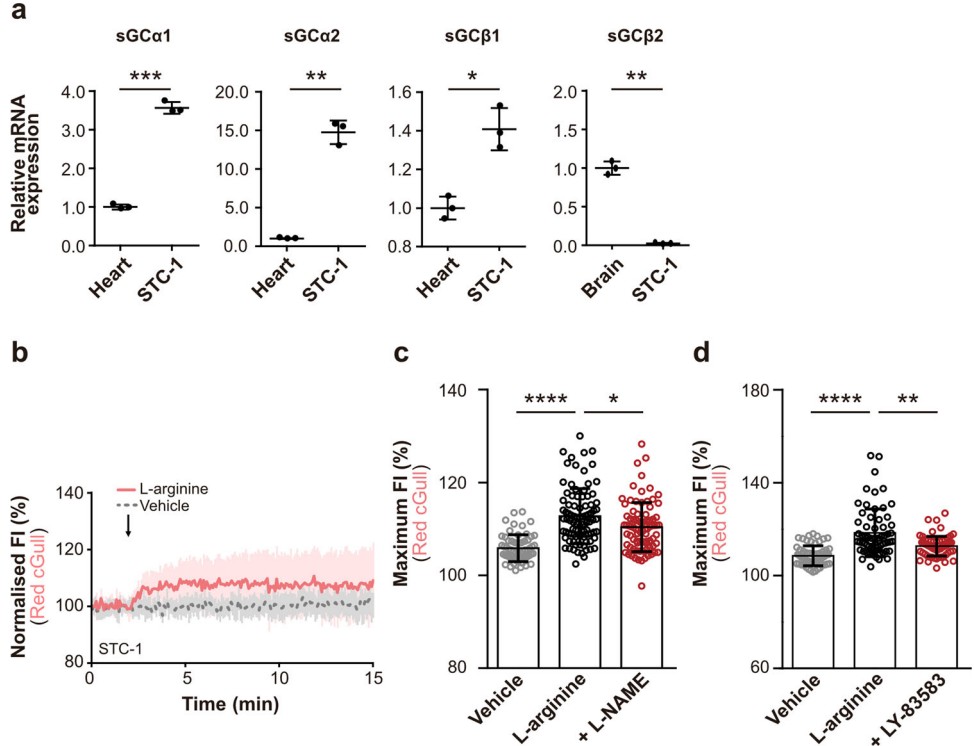

**Fig. 5 L-arginine potentiates cGMP production in STC-1 cells via NO-cGMP signalling. a** Relative mRNA expression level of cytosolic/soluble guanylyl cyclase (sGC) subunits (α1, α2, β1, and β2) in STC-1 cells measured by real-time PCR. GAPDH was used as the internal control. Data are means ± standard deviation (s.d.) from three independent experiments. Welch's *t*-test. From left to right: $p = 0.0003$ (***), $p = 0.004$ (**), $p = 0.0102$ (*), $p = 0.0024$ (**) (two-tailed). **b** Time course of FI from Red cGull-expressing STC-1 cells during the application of 10 mM L-arginine (solid line) or vehicle (dashed line). Data were means ± s.d. (L-arginine: $n = 60$ cells from four independent experiments; vehicle: $n = 65$ cells from four independent experiments). **c** Maximum FIs of Red cGull during the application of 10 mM L-arginine with 5 mM L-NAME (NOS inhibitor). Data were means ± s.d. (vehicle: $n = 60$ cells from four independent experiments; L-arginine: $n = 89$ cells from five independent experiments, +L-NAME: $n = 84$ from five independent experiments). Kruskal–Wallis test with Dunn's multiple comparisons. From left to right: $p < 0.0001$ (****), $p = 0.0419$ (*). **d** Maximum FIs of Red cGull during the application of 10 mM L-arginine with 1 μM LY-83583 (sGC inhibitor). Data were means ± s.d. (Vehicle and L-arginine: same data as in **b**, +LY-83583: $n = 71$ cells from four independent experiments). Kruskal–Wallis test with Dunn's multiple comparisons. From left to right: $p < 0.0001$ (****), $p = 0.0066$ (**).

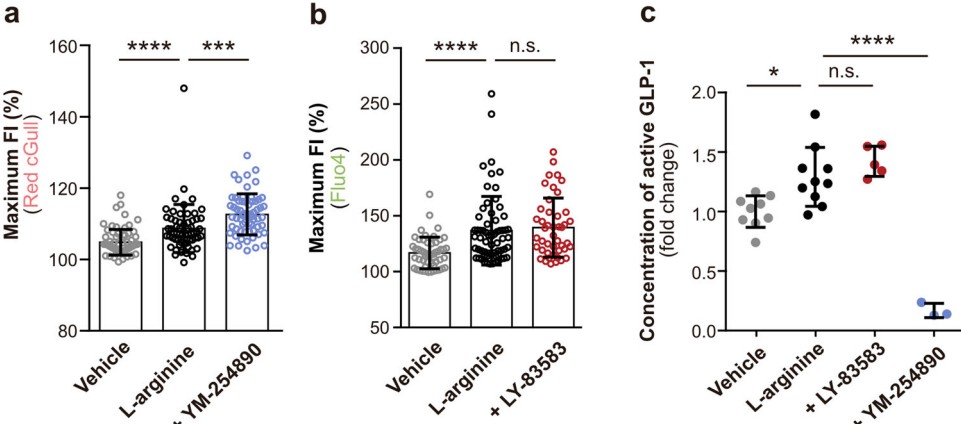

**Fig. 6 Interplay between Ca²⁺ and cGMP, and their effect on GLP-1 secretion in STC-1 cells. a** Maximum FI of Red cGull with the application of 10 mM L-arginine in the presence of 250 nM YM-254890. Data were means ± s.d. (vehicle: $n = 72$ cells from four independent experiments; L-arginine: $n = 60$ cells from four independent experiments; +YM-254890: $n = 62$ from four independent experiments). Kruskal–Wallis test with Dunn's multiple comparisons. From left to right: $p < 0.0001$ (****), $p = 0.0005$ (***). **b** Maximum FI of Fluo4 during the application of 10 mM L-arginine in the presence of 1 μM LY-83583. Data were means ± s.d. (vehicle: $n = 51$ cells from four independent experiments, L-arginine: $n = 64$ cells from four independent experiments; +LY-83583: $n = 45$ cells from four independent experiments). Kruskal–Wallis test with Dunn's multiple comparisons. From left to right: $p < 0.0001$ (****), $p > 0.9999$ (n.s.). **c** Results of ELISA showing the concentration of GLP-1 secreted by STC-1 cells after the application of 10 mM L-arginine in the absence or presence of each inhibitor. Data were means ± s.d. One-way ANOVA with Tukey's multiple comparison test (vehicle: $n = 9$ wells, L-arginine: $n = 10$ wells, +LY-83583: $n = 5$ wells, +YM-254890: $n = 3$ wells). From left to right: $p = 0.0103$ (*), $p = 0.5639$ (n.s.), $p < 0.0001$ (****).

The response of Red cGull was 7.7 ($F/F_0$) at the end of the site-directed random mutation screening process (Supplementary Fig. 1), which was higher than the response of Red cGull after purification (Fig. 2a). We think that the response of the purified protein was more reliable because they are free of potential contaminants. Thus, we finally defined the dynamic range of Red cGull as 6.7 ($F/F_0$) according to the emission spectra of purified recombinant protein data (Fig. 2a). It has been reported that, in single red FP-based indicators based on mApple, such as R-GECO, the protonated chromophore without fluorescence absorbs near 450 nm and the deprotonated chromophore with fluorescence absorbs near 570 nm[32]. Moreover, we found that change in quantum yield after the application of cGMP. Therefore, both changes respond to cGMP in Red cGull induces its FI increase. We found that the $EC_{50}$ of Red cGull was 0.33 µM, which was close to the widely used FRET-based cGMP indicator[15]. The physiological level of cGMP is believed to be <5 µM; thus, Red cGull should be applicable at the physiological cGMP level in many cells[15]. However, because certain cells, such as cardiomyocytes or stellate ganglion neurons, contain low cGMP concentrations (nM range), more sensitive indicators with higher affinities to cGMP might be required to expand the usability of the indicator[33].

To examine the utility of Red cGull in live-cell imaging, we expressed Red cGull in various cell lines. Both the NO donor SNAP and the cGMP analogue 8-Br-cGMP induced an increase in FI; however, their kinetics seemed to be different. This difference would be due to the difference in the properties of those compounds. 8-Br-cGMP is a phosphodiesterase-resistant cGMP analogue and showed monotonically increasing cGMP changes over time, whereas SNAP produced cGMP via the activation of sGC; the produced cGMP was degraded by phosphodiesterase. During dual-colour imaging in GLUTag cells, SNAP triggered increases in both $[Ca^{2+}]_i$ and $[cGMP]_i$. Such SNAP-dependent $[cGMP]_i$ and $[Ca^{2+}]_i$ increase may reflect the cGMP-dependent activation of cyclic nucleotide-gated channels, which is one signalling pathway previously reported in relation to a cGMP-dependent $[Ca^{2+}]_i$ increase[34]. The FI of Red cGull was saturated at ~2.5-fold in live-cell imaging, which is lower than its dynamic range (6.7 ($F/F_0$)) obtained via in vitro spectrometry with purified Red cGull proteins. This difference may be explained by a certain basal level of cGMP in cells[15].

The ability to monitor and manipulate intended signal transduction is becoming essential to the study of cell biology and especially neuroscience. By using Red cGull and photoactivated guanylyl cyclase, i.e. BlgC, we were able to manipulate cGMP production and then monitor cGMP in real-time. With repeated blue light stimuli, we showed that only a short duration of exposure (0.2 s) was sufficient to produce cGMP. We believe that longer stimulation would produce more cGMP, which may degrade quickly after they are produced; hence, the responses appear to be similar to that of short exposures. We think that the combination of cGMP imaging and optogenetic techniques has the potential to expand the field and help provide answers to important questions.

L-arginine, a conditionally essential amino acid, is contained in various foods and produced in cellular metabolism. L-arginine is transported via cationic amino acid transporters and used to produce NO, which in turn promotes the generation of cGMP by sGC[35]. sGC is a heterodimer with α and β subunits. Within two isoforms, α1 and β1 are physiologically functional in most tissues, including the small intestine[27]. Although our data show that L-arginine potentiates GLP-1 secretion in STC-1 cells, which is consistent with previous in vivo experiments, the cGMP produced by L-arginine application itself did not stimulate GLP-1 secretion[12,13]. In pancreatic β cells, cGMP derived from NO

stimulates insulin secretion; however, this effect reportedly occurs only when NO exists in low concentrations (<50 nM); when NO concentrations are high (>80 nM), insulin secretion is instead somewhat inhibited[6]. Because a bidirectional effect of NO exists, 10 mM of L-arginine might not be a suitable concentration to activate the NO/cGMP signalling pathway and induce hormone secretion in STC-1 cells. L-arginine is also sensed by all three promiscuous amino acid sensing receptors. In our work, only a CaSR antagonist, Calhex231, suppressed the L-arginine-induced $Ca^{2+}$ increase; however, it was reported that GPRC6A antagonist, Calindol, also functions as an allosteric modulator of CaSR[36]. Therefore, we consider whether the inhibitory effect on GPRC6A or activation effect on CaSR may have been counterbalanced and whether Calindol had no effect on $Ca^{2+}$ signalling. Thus, it is difficult to judge whether GPRC6A is relevant in L-arginine signalling. Clemmesen et al. used GPRC6A-knockout mice and determined that GPRC6A is not required in L-arginine-mediated GLP-1 secretion[37]. Further experiments, such as those including specifically targeted antagonists or knockout mice, will be required to reveal the mechanism of L-arginine-mediated GLP-1 secretion in the future.

## Conclusion

In conclusion, we successfully developed the red single FP-based cGMP indicator, which we have named Red cGull. In vitro analysis showed that the FI of Red cGull increased more than sixfold in the presence of cGMP. Red cGull is applicable for live-cell imaging of cGMP, including dual cGMP/$Ca^{2+}$ imaging and for cGMP imaging in conjunction with optogenetic stimulation. Overall, our results demonstrate that Red cGull overcomes the limited utility of FRET-based or green FP-based indicators to accommodate existing excitation and emission wavelength windows. The accessibility of our technology will enable researchers in a range of disciplines to investigate the NO/cGMP signalling pathway, e.g., in health and disease, in not only enteroendocrine cells but also other cells, tissues and animal models.

## Methods

**Chemicals.** Imidazole was purchased from Tokyo Chemical Industry (Tokyo, Japan). Nω-Nitro-L-arginine methyl ester hydrochloride (L-NAME), 8-bromoguanosine 3′, 5′-cyclic monophosphate sodium salt (8-Br-cGMP) and cGMP, and cyclic adenosine 3′, 5′-monophosphate (cAMP) were purchased from Sigma-Aldrich (St. Louis, MO, USA). S-nitroso-N-acetylpenicillamine (SNAP) was purchased from Cayman Chemical (Ann Arbor, MI, USA). L-arginine was purchased from Nacalai Tesque, Inc. (Kyoto, Japan). LY-83583 and YM-254890 were purchased from FUJIFILM Wako Pure Chemical Corporation (Osaka, Japan). Calindol-13C, d2 Hydrochloride (Calindol) was purchased from Santa Cruz Biotechnology (Dallas, TX, USA). Calhex231 was purchased from Abcam (Cambridge, UK).

**Plasmid construction.** The DNA fragment of a red FP variant, mApple, was created by DNA synthesis from Integrated DNA Technologies (Coralville, IA, USA).[29] Subsequently, mApple was modified by PCR to insert SacII and EcoRI restriction enzyme sites at a position between A150 and V151 before being cloned into the pRSET-A vector at the BamHI/HindIII site, as previously described[17]. During this step, the hyper acidic region of a fragment of mouse amyloid precursor protein (APP; NM_001198823.1, amino acids 190–286) was amplified from adult mouse whole brain mRNA via RT-PCR and then fused to the N-terminus of mApple to improve the solubility of the protein under bacterial expression[38]. The cDNA for the cGMP-binding GAFa domain of mouse phosphodiesterase 5α (mPDE5α, NM_153422.2, amino acids 164–298) was amplified from adult mouse whole brain mRNA using RT-PCR and then inserted into the SacII/EcoRI sites of the modified mApple in pRSET-A (i.e. the Red cGull prototype)[15,24]. To expand the dynamic range of the indicator by linker length optimisation, leucine zipper sequences of various lengths were inserted between mApple and the cGMP-binding domain[39]. The mutant with the greatest fluorescence intensity was selected for further optimisation. PCR was performed using two sets of primers, including NNK and MNN to introduce random mutations at one position in the linker amino acid sequences. In one screening, ~50 random mutations were produced; the mutation with the greatest dynamic range was selected as the next template for site-directed random PCR. After the repetitive screening, a mutant that resulted in the

maximum dynamic range by cGMP was produced; this mutant was named Red cGull. For expression in mammalian cells, Red cGull was subcloned into the pcDNA3.1(−) vector (Thermo Fisher Scientific, Waltham, MA, USA) at the BamHI/HindIII site. To generate photoactivated guanylyl cyclase (i.e. BlgC), the DNA sequence of bPAC was amplified by PCR from pGEM-HE-h_bPAC_cmyc (Addgene, Watertown, MA, USA; #28134) and three point mutations (i.e. K197E, D265K and T267G) were introduced to facilitate guanylyl cyclase activity based on a previous study[21]. The resultant sequence was inserted into the pEGFP-C1 vector for live-cell imaging. To generate lentivirus, Red cGull was subcloned into the backbone vector plasmid, CSII-EF-MCS (RIKEN RBC DNA BANK, #RDB04378), at the NotI/BamHI site.

**Protein expression and purification.** For protein expression, pRSET-A with APP-fused Red cGull was transformed into *Escherichia coli* JM109 (DE3) (Promega, Madison, WI, USA) and cultured in LB medium with 50 mg/L ampicillin (FUJI-FILM Wako Pure Chemical Corporation) at 20 °C for 4 days. The cells were then centrifuged at 7000 rpm and 4 °C for 10 min. The resultant pellets were suspended in phosphate-buffered saline (PBS) with 40 µg/mL lysozyme (FUJIFILM Wako Pure Chemical Corporation) and lysed by freeze-thawing and ultrasonic homogenisation. Subsequently, nickel-nitrilotriacetic acid agarose beads (QIAGEN, Venlo, Netherlands) were added to the recovered supernatant and absorbed via rotation at 4 °C for 3–6 h. The beads were then recovered by centrifugation and resuspended in PBS. The supernatant was added to a filtered column, washed three times with PBS, washed three times with 3 mL of 10 mM imidazole/PBS, and then eluted using 5 mL of 300 mM imidazole/PBS. To remove imidazole, 1 mL of elution was added to a PD-10 filtration column (GE Healthcare, Buckinghamshire, UK) in HEPES buffer (150 mM KCl and 50 mM HEPES). This purified protein was analysed to measure excitation and emission spectra, a dose-response curve, and absorption spectra.

**In vitro spectrometry.** Excitation (for 595 nm) and emission (at 550 nm) spectra were measured in the absence or presence of 10 µM cGMP in 5 µM of purified protein/HEPES buffer using a fluorescence spectrophotometer (F-2500; Hitachi, Tokyo, Japan). The absorption spectra of Red cGull were measured using a UV spectrometer (UV-1800; Shimadzu, Kyoto, Japan). Absorption spectra was corrected using the absorbance at 650 nm as the background. The quantum yields were measured by the absolute photoluminescence quantum yield measurement system (Hamamatsu Photonics K.K., Shizuoka, Japan, C9920-02: excitation wavelength: 550 nm). To generate a dose-response curve, the fluorescence intensity (FI) of the purified proteins, diluted to 1 µM with HEPES, was measured in the presence or absence of cGMP or cAMP. The $EC_{50}$ was calculated using the equation for the four-parameter logistics curve in the Rodbard mode of ImageJ's curve fitter function (National Institutes of Health, Bethesda, MD, USA). The normalised dynamic range for cAMP was converted based on the maximum and minimum parameters of cGMP measurements; a logarithmic approximation line was then drawn.

**Cell culture.** HeLa, HEK293T (AAV pro®293T cell line, TAKARA BIO Inc., Shiga, Japan), GLUTag (kindly provided by Dr. Daniel Drucker, The Lunenfeld-Tanenbaum Research Institute), and STC-1 (purchased from ATCC, Manassas, VA, USA; ATCC® CRL-3254™) cells were cultured in Dulbecco's modified Eagle's medium (DMEM; Sigma-Aldrich; high glucose: HeLa, HEK293T and STC-1; low glucose: GLUTag) supplemented with L-glutamine, sodium pyruvate, 10% (v/v) heat-inactivated foetal bovine serum (Sigma-Aldrich), and 100 U/mL penicillin with 100 µg/mL streptomycin (Sigma-Aldrich) at 37 °C and in a 5% $CO_2$ atmosphere.

**Lentivirus production.** For lentivirus production, the following three transfection plasmids were mixed in 15 mL tubes: 10 µg of pCAG-HIVgp (RIKEN BRC DNA BANK, #RDB04394), 10 µg of pCMV-VSV-G-RSV-Rev (RIKEN RBC DNA BANK, #RDB04393), and 17 µg of CSII-EF-Red cGull. DNA mixtures were then diluted with 36 µL of Lipofectamine 2000 Transfection Reagent (Thermo Fisher Scientific) and incubated for 20 min at room temperature. HEK293T cells were seeded onto 10 cm culture dishes at a density of $5 \times 10^5$ cells and then the transfection mixture was transferred to the cells before they were incubated at 37 °C under a 5% $CO_2$ atmosphere. Viruses were harvested at 48 and 96 h post-transfection. Viral supernatants were centrifuged at 780×g and 4 °C for 10 min and then the supernatant was filtered through a 0.45 µm filter (Merck Millipore, Burlington, MA, USA) to recover the virus.

**Plasmid transfection and lentivirus infection.** For live-cell imaging, cells were trypsinised and plated onto poly-L-lysine (PLL; Sigma-Aldrich)-coated glass coverslips in 35-mm dishes. Two days after plating, the cells were transfected with plasmids (1.5 µg for single-wavelength imaging and 0.5 µg each for two-wavelength imaging with photoactivated protein) using 3 µL of Lipofectamine 2000 Transfection Reagent (Thermo Fisher Scientific) according to the manufacturer's protocol. Four hours after transfection, the medium was changed, and the cells were

cultured at 30–32 °C for 2 days until imaging was conducted. For lentivirus infection, STC-1 cells were plated in PLL-coated dishes (35 mm). Two days after plating, 1 mL of lentivirus and 25 µg/mg of polybrene (Nacalai Tesque) were mixed and added to the cell culture with 1 mL of DMEM (high glucose). The cells were then incubated at 37 °C under a 5% $CO_2$ atmosphere for 2 days. On the day before the imaging experiment, the high-glucose DMEM culture medium was changed to a low-glucose DMEM medium, and cells were cultured at 30–32 °C.

**Live-cell imaging.** Experimental procedures were mainly based on our previous work[18,19]. Briefly, cells plated onto 35-mm dishes were washed and imaged in modified Ringer's buffer (RB; 140 mM NaCl, 3.5 mM KCl, 0.5 mM $NaH_2PO_4$, 0.5 mM $MgSO_4$, 1.5 mM $CaCl_2$, 10 mM HEPES, and 2 mM $NaHCO_3$ [pH 7.4]). Fluo4-AM (Dojindo Laboratories, Kumamoto, Japan) was loaded for GLUTag cells for 20 min at 37 °C prior to conducting imaging experiments. Single fluorescence imaging (GLUTag and HeLa cells) was performed using an inverted microscope (IX-71; Olympus, Tokyo, Japan) equipped with an oil-immersion objective lens (UApo/340, 40×, NA = 1.35; Olympus), intermediate magnification lens (×1.6; Olympus), and EM-CCD camera (Evolve; Photometrics, Tucson, AZ, USA), for which the exposure was controlled by MetaMorph software (Molecular Devices, Sunnyvale, CA, USA). Images were acquired using a xenon lamp (U-LH75XEAPO; Olympus), 545–580-nm excitation filter, 585-nm dichroic mirror, and 610-nm emission filter (UWIY2; Olympus). SNAP or 8-Br-cGMP was applied manually 2 min after the beginning of imaging acquisition. For dual-colour imaging, photoactivation imaging, and single fluorescence imaging of STC-1, imaging was performed using an inverted microscope (Axio Observer D1; Carl Zeiss, Oberkochen, Germany) equipped with an oil-immersion objective lens (UPlanApo, 40×, NA = 1.00; Olympus) and CMOS camera (ORCA-Flash4.0v2, C11440; Hamamatsu Photonics K.K.), for which the exposure was controlled by MetaMorph software (Molecular Devices). Fluorescence images were acquired using a mercury lamp (HBO100; Carl Zeiss), with a BP470/40 and FT495 filter (B38-HE; Carl Zeiss) for Fluo4 and a BP530-585 and FT600 filter (G00; Carl Zeiss) for Red cGull. During photoactivation imaging, blue light was stimulated at the following imaging planes and lengths; 150 (0.2 s), 300 (1 s), 450 (20 s) and 600 (60 s). L-NAME was dissolved in deionized water and preincubated for 30 min at 37 °C. LY-83583 was dissolved in DMSO and preincubated for 30 min at 37 °C. YM-254890, Calhex231 and Calindol were first dissolved in DMSO and then applied 2 min after the beginning of imaging acquisition. MetaMorph was used to analyse the acquired images, including quantification of cell fluorescence intensity. After subtracting the background, FIs normalised to 100% were calculated as average FIs between 90 and 120 sec from the beginning of imaging acquisition.

**RNA isolation and real-time PCR.** Total RNA from STC-1 cells and mouse tissues (heart, brain, and testis) was isolated using RNeasy Mini Kit (QIAGEN). After DNase treatment using RNase-Free DNase Set (QIAGEN), cDNA was synthesised using High-Capacity RNA-to-cDNA Kit (Thermo Fisher Scientific). Synthesised cDNA was amplified using THUNDERBIRD SYBR qPCR Mix (TOYOBO, Osaka, Japan). The reaction was performed using Thermal Cycler Dice (TAKARA BIO Inc.) in a two-step reaction (95 °C for 1 min for initial denaturation, 95 °C 15 s, 60 °C 30 s; 40 cycles). Primer sequences used in real-time PCR analysis are shown in Supplementary Table 1. The experiments were repeated three times. GAPDH was used as the reference gene in each template, and the relative expression level of the target gene was calculated using the ΔΔCt method. The expression level of the tissue was then used as a positive control, normalised to 1 for each gene, and the expression levels of the STC-1 cells were calculated.

**Enzyme-linked immunosorbent assay.** STC-1 cells were plated in 24 well plates at $1.0 \times 10^5$ cells per well. After 24 h, the cultured medium was changed from high- to low-glucose DMEM. Two days after plating, cells were washed twice with RB containing 2.2 mM glucose and 0.5% (w/v) bovine serum albumin (BSA; FUJIFILM Wako Pure Chemical Institute). Subsequently, vehicle solution (dimethyl sulfoxide), 10 mM L-arginine, or 10 mM L-arginine with either 1 µM LY-83583 or 250 nM YM-254890 in RB containing 2.2 mM glucose and 0.5% BSA were applied to the cells and incubated for 2 h at 37 °C under a 5% $CO_2$ atmosphere. After incubation, cell supernatant was collected into collection tubes supplemented with 60-KIU aprotinin (FUJIFILM Wako Pure Chemical Institute) and 34 µg/mL diprotin A (Peptide Institute, Inc., Osaka, Japan). After centrifugation at 1000×g and 4 °C for 10 min, the supernatant was diluted in RB containing 2.2 mM glucose and 0.5% (w/v) BSA and then used for analysis with a GLP-1 Active Form Assay Kit (#27784; IBL, Gunma, Japan) and microplate reader (Varioskan LUX; Thermo Fisher Scientific) according to the manufacturer's protocols. Changes were calculated relative to vehicle-treated GLP-1 secretion.

**Statistics and reproducibility.** Final data are shown as means ± standard deviations from $n$ cells of more than three independent experiments, as indicated in the figure legends. When using bar charts, individual data points were overlaid. $p$ value of 0.05 was considered significant. $p$ values were stated in the graph legend. Statistical analysis in real-time PCR was performed using a two-tailed Welch's t-test. Normality was evaluated by Shapiro–Wilk test, and a multiple comparison test,

whether parametric or nonparametric, was selected based on the normality test. Statistical analysis in imaging acquisition were performed using Kruskal–Wallis test with Dunn's multiple comparison. Statistical analysis in ELISA was performed using a One-way ANOVA with Tukey's multiple comparison test. These analyses were conducted in GraphPad Prism 6 software (GraphPad Software, La Jolla, CA, USA).

**Reporting summary**. Further information on experimental design is available in the Nature Research Reporting Summary linked to this paper.

## Data availability

The data supporting the findings of this study are available within the article and its Supplementary Information. Plasmids generated in this study have been deposited to Addgene. The source data for the main figures and extended data figures are available from the corresponding authors on request.

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

## Acknowledgements

The authors would like to thank Dr. Daniel Drucker for kindly providing GLUTag cells, Dr. Keitaro Yoshimoto for technical assistance and Enago (www.enago.jp) for their English language review. This work was supported by the Japan Society for the Promotion of Science Grant-in-Aid for Scientific Research (KAKENHI) (JP20J14411 to M.M.; JP16J06838, JP19K23820 and JP20K16118 to K.H.; JP17K08529, JP20H00575, JP20H04121, JP20H04765 and JP20H04836 to T.T.; JP16K01922 and JP18H04832 to T.K.), Kowa Life Science Foundation (to K.H.), the Uehara Memorial Foundation (to K.H.), Research Foundation for Opto-Science and Technology (to K.H.), and the Yamaguchi Educational and Scholarship Foundation (to T.K.). The authors also thank Dr. Kimihisa Yamamoto and Dr. Tetsuya Kambe (absolute photoluminescence quantum yield measurement system) for technical assistance.

## Author contributions

R.I. performed experiments on sensor developing screening and analysed data. M.T., Y.O. and R.I. performed the experiments, including cell culture and imaging. M.T. analysed data and prepared figures and manuscripts. M.M. and K.H. performed qRT-PCR and provided technical support on imaging acquisition. H.U., T.K. and T.T. designed experiments and made final edits. All authors discussed the results and prepared the paper.

## Competing interests

The authors declare no competing interests.
