## [Peer Review File · Communications Biology]

Reviewers' comments:

Reviewer #1 (Remarks to the Author):

This paper describes the important and interesting development of a cGMP-sensing red fluorescent protein, potentially the first of its kind. The paper is well-written but a bit more concise that I would like with describing the supporting data.

I have two major concerns.

1. I do not find the data presented in Figs. 5-6 for the increased fluorescence with arginine and the effects of L-NAME, LY-83583, and YM-254890 convincing. The confidence intervals for the different categories of data overlap to a great extent. The data for L-arginine treatment have a large variance. The data actually support no statistically significant differences in fluorescence. The magnitude of the observed differences are also fairly small, ~50%.
2. The concentrations of some of the reagents used, such as SNAP, 8-Br-cGMP, are very high (>500 μ M) and non-physiological. Use of reagents at such high concentrations can lead to non-specific biological effects. The authors should justify the use of such high concentrations. Ideally, the authors should demonstrate dose-dependent effects.

Reviewer #2 (Remarks to the Author):

This manuscript describes a design and a proof of function of a new genetically encoded sensor for cGMP, Red cGull. This is a new red FP sensor based on a single circularly permuted FP. Red shifted sensors can provide additional colors in bioimaging and can be used simultaneously with optogenetic tools that usually absorb in blue-green region of spectrum. The authors experimentally demonstrate these advantages of Red cGull. The sensor will certainly find applications in a wide field of cell physiology. The paper can be published after minor corrections.

1. The references 25 and 26 are not very relevant to the discussion in lines 90-95, because they are not considering different protonation states of circularly permuted mApple. The most relevant reference is Molina, R. S. et al. (2019), Understanding the Fluorescence Change in Red Genetically Encoded Calcium Ion Indicators. *Biophysical Journal*, 116(10), 1873–1886, where it is shown that in circularly-permuted red FPs, based on mApple (RGECO family), the neutral chromophore absorbs near 450 nm and the anionic chromophore - near 570 nm. It is also shown there that the molecular mechanism of fluorescence change involves, among other factors, deprotonation of chromophore.
2. In Fig. 2b, there is no peak at 420 nm (line 89 of the manuscript), but there is one at ~450 nm, that corresponds to a neutral chromophore, as demonstrated in the above paper. That should be corrected.
3. As already mentioned, reference 25 does not present a peak at 400 nm in the mApple absorption spectrum, because at the conditions of reference 25 neutral chromophore is not present.
4. The protonated/deprotonated absorption ratio of Red cGull can be estimated from Fig 2 as ~1.6 only near 570 nm. How can the authors explain the 6.6 times change of fluorescence upon saturation with cGMP? Does the quantum yield of fluorescence also change?

[Response to Reviewer #1]

We really appreciate your comments. We would like to respond to your comments point by point as follows:

1. I do not find the data presented in Figs. 5-6 for the increased fluorescence with arginine and the effects of L-NAME, LY-83583, and YM-254890 convincing. The confidence intervals for the different categories of data overlap to a great extent. The data for L-arginine treatment have a large variance. The data actually support no statistically significant differences in fluorescence. The magnitude of the observed differences are also fairly small, ~50%.

We do understand your comment. We admit that the increase in fluorescence intensity by L-arginine is not quite big and has a large variance. However, for statistical analysis, we have already analyzed every data according to the statistical guidelines of the *Communications Biology*. Therefore, we have additionally included statistical information in method section, added means \pm standard deviation in the main text, and wrote actual p value on each graph legend.

For large variance of response, we believe it is common for bioimaging using cell culture under fluorescence microscope, because cell response is often affected by phase of cell cycle, which is usually distinct between each cell. Also, when we apply a reagent to dishes, it is experimentally difficult to stimulate all cells with the same concentration simultaneously.

To confirm the reliability of the increase of cGMP by L-arginine, we conducted additional experiments using δ -FlnG, which is a previously reported green fluorescent protein-based cGMP indicator (Nausch, L. W. M. et al. Differential patterning of cGMP in vascular smooth muscle cells revealed by single GFP-linked biosensors. *Proc. Natl. Acad. Sci. USA* 105, 365–370 (2008)). First, we applied SNAP to δ -FlnG -expressing STC-1 cells and found the increase of fluorescence intensity (Supplementary Fig. 6a, c). Next, we observed small, but statistically significant increase of fluorescence intensity in δ -FlnG by application of L-arginine in STC-1 cells as well (means \pm s.d., 118.5 ± 12.8 , $p < 0.0001$, Supplementary Fig. 6b, c).

We added explanations of this experiment in the manuscript as follows: “For validation of those response, we used Red cGull nega (Supplementary Fig. 5 c, d), and δ -FlnG...cGMP indicator (means \pm standard deviation (s.d.), 113.7 ± 8.3 , Supplementary Fig. 6a, c)²⁰.” (Page 6, line 118), and “We also applied L-arginine...as well (means \pm s.d., 118.5 ± 12.8 , $p < 0.0001$, Supplementary Fig. 6b, c)²⁰.” (Page 7, line 150).

For those reasons, we believe that Red cGull detects the increase of cGMP induced by L-arginine treatment in STC-1 cells.

2. The concentrations of some of the reagents used, such as SNAP, 8-Br-cGMP, are very high (>500 μ M) and non-physiological. Use of reagents at such high concentrations can lead to non-specific biological effects. The authors should justify the use of such high concentrations. Ideally,

the authors should demonstrate dose-dependent effects.

We appreciate your insightful suggestion. We applied low dose of SNAP and 8-Br-cGMP to cells and monitored the fluorescence intensity of Red cGull. For SNAP application, we applied 6 to 600 μM SNAP to GLUTag cells. 6 μM SNAP induced small increase of fluorescence intensity, and 60 μM SNAP induced large increase of fluorescent intensity as 600 μM SNAP but decreased over time. For 8-Br cGMP application, we applied 0.5 to 2 mM 8-Br-cGMP to HeLa cells. 0.5 mM 8-Br-cGMP did not induced changes in fluorescence intensity, weather 1.0 mM induced comparable increase as 2 mM 8-Br-cGMP.

We added explanations of this experiment in the manuscript as follows: “Application of various concentration of SNAP ... increases as 2 mM 8-Br-cGMP.” (Page 6, line 112).

Additionally, we conducted those imaging experiments by using Red cGull nega, which showed little change of fluorescence intensity in the presence of cGMP. Neither 600 μM SNAP nor 2 mM 8-Br cGMP increased fluorescence intensity after application, suggesting that the response shown in Red cGull was not including non-physiological effect.

We added explanations of this experiment in the manuscript as follows: “For validation of those response, we used Red cGull nega (Supplementary Fig. 5c, d) ... cGMP indicator (means \pm standard deviation (s.d.), 113.7 ± 8.3 , Supplementary Fig. 6a, c)²⁰.” (Page 6, line 118).

[Response to Reviewer #2]

We really appreciate your comments. We would like to response to your comments point by points as follows:

1. *The references 25 and 26 are not very relevant to the discussion in lines 90-95, because they are not considering different protonation states of circularly permuted mApple. The most relevant reference is Molina, R. S. et al. (2019), Understanding the Fluorescence Change in Red Genetically Encoded Calcium Ion Indicators. Biophysical Journal, 116(10), 1873–1886, where it is shown that in circularly-permuted red FPs, based on mApple (RGECO family),*

We apologize for inappropriate references, and appreciate your insightful comments and suggestion for the most relevant reference. We removed the previous references #25 and #26, and add a new reference #33 (Molina, R. S. *et al.*) to the manuscript.

the neutral chromophore absorbs near 450 nm and the anionic chromophore - near 570 nm. It is also shown there that the molecular mechanism of fluorescence change involves, among other factors, deprotonation of chromophore.

We appreciate your comments. We corrected the manuscript and included it in discussion section as follows: “It has been reported that, in single red FP-based indicators based on mApple, such as R-GECO, the protonated chromophore without fluorescence absorbs near 450 nm and the deprotonated chromophore with fluorescence absorbs near 570 nm³³.” (Page 8, line

194).

We also included a description about other factors involving molecular mechanism of fluorescence intensity of Red cGull in response to your comment #4.

2. In Fig. 2b, there is no peak at 420 nm (line 89 of the manuscript), but there is one at ~450 nm, that corresponds to a neutral chromophore, as demonstrated in the above paper. That should be corrected.

We apologize for incorrect description. As you pointed out, there is a peak at ~450 nm instead of ~420 nm. We corrected the manuscript as follows: “The absorption spectra of Red cGull showed a higher peak near 450 nm in absence of cGMP, and a higher peak near 570 nm in the presence of 100 μ M cGMP (Fig. 2b).” (Page 5, line 92).

3. As already mentioned, reference 25 does not present a peak at 400 nm in the mApple absorption spectrum, because at the conditions of reference 25 neutral chromophore is not present.

We apologize for incorrect description. We removed reference #25 and corrected the manuscript in reference to Molina, R. S. et al. (2019) as follows: “It has been reported that, in single red FP-based indicators based on mApple, such as R-GECO, the protonated chromophore without fluorescence absorbs near 450 nm and the deprotonated chromophore with fluorescence absorbs near 570 nm³³.” (Page 8, line 194).

4. The protonated/deprotonated absorption ratio of Red cGull can be estimated from Fig 2 as ~1.6 only near 570 nm. How can the authors explain the 6.6 times change of fluorescence upon saturation with cGMP? Does the quantum yield of fluorescence also change?

We appreciate your insightful comments. To assess the brightness of Red cGull, we measured the quantum yield and calculated extinction coefficients in the cGMP-free and cGMP-present states. We compared those values with those of Red cGull nega which shows little change in fluorescence intensity in the presence of cGMP. We added the manuscript on these data as follows: “Extinction coefficients were calculated ...and extinction coefficient and quantum yield were displayed in Table 1” (Page 5, line 94).

We discussed on this in discussion section as follows; “It has been reported that,...suggesting that other mechanisms such as excited-state proton transfer would be involved in the fluorescence intensity change.” (Page 8, line 194).

Reviewers' comments:

Reviewer #1 (Remarks to the Author):

Thank you for carefully considering the comments of the Reviewers. The paper is much improved.

Reviewer #2 (Remarks to the Author):

Most of my questions were addressed. However question 4 was not directly addressed. It looks like the absorption spectra shown in Fig. 2b contain a background that is not due to absorption, but to reflectance and/or scattering. Therefore the peak optical density of the samples can be overestimated. It is difficult to estimate the background value from the data in Fig. 2b, because the spectra were not recorded above 600 nm. The raw absorbance measured at 610 - 650 nm region, where the spectrum should look flat, could provide these background values. Then the background should be subtracted to obtain the real absorption peak values.

This should be corrected, and this correction could potentially explain the 6-fold enhancement of fluorescence signal, because right now the estimated enhancement is $0.225 \times 7394 / (0.149 \times 4050) = 2.76$ only.

The paper can be accepted after this correction.

[Response to Reviewer #1]

Thank you for carefully considering the comments of the Reviewers. The paper is much improved.

We greatly appreciate reviewer's insightful comments for improving our manuscript.

[Response to Reviewer #2]

Most of my questions were addressed. However, question 4 was not directly addressed. It looks like the absorption spectra shown in Fig. 2b contain a background that is not due to absorption, but to reflectance and/or scattering. Therefore, the peak optical density of the samples can be overestimated. It is difficult to estimate the background value from the data in Fig. 2b, because the spectra were not recorded above 600 nm. The raw absorbance measured at 610 - 650 nm region, where the spectrum should look flat, could provide these background values. Then the background should be subtracted to obtain the real absorption peak values. This should be corrected, and this correction could potentially explain the 6-fold enhancement of fluorescence signal, because right now the estimated enhancement is $0.225 \times 7394 / (0.149 \times 4050) = 2.76$ only. The paper can be accepted after this correction.

We appreciate the reviewer's insightful suggestion. We measured the absorption spectra up to 650 nm again, and corrected the background using absorbance at 650 nm as reviewer suggested. Then, previous Table 1 and Figure 2b were replaced with new ones, and added the explanation of background correction in method section. In addition to an emphasized change in absorbance by this background correction, the spectra acquired in this revision show noticeable peaks corresponding to protonated/deprotonated forms compared to the previous spectra, which would be due to the differences in maturation of chromophore in purified indicator proteins by the different storage period. After the calculation of extinction coefficient, we found that the estimated enhancement is $(0.225 \times 5,638) / (0.149 \times 988) = 8.6$. We believe this value is close to what reviewer expected. Since the quantum yield and extinction coefficient can explain the dynamic range of Red cGull, we refrained from referring to proton transfer in discussion section.

REVIEWERS' COMMENTS:

Reviewer #2 (Remarks to the Author):

All my questions and comments were addressed and the manuscript can be published now.